# Vision-Language Models Provide Promptable Representations for Reinforcement Learning

## Abstract

Humans can quickly learn new behaviors by leveraging background world knowledge. In contrast, agents trained with reinforcement learning (RL) typically learn behaviors from scratch. We thus propose a novel approach that uses the vast amounts of general and indexable world knowledge encoded in vision-language models (VLMs) pre-trained on Internet-scale data for embodied RL. We initialize policies with VLMs by using them as promptable representations: embeddings that encode semantic features of visual observations based on the VLM's internal knowledge and reasoning capabilities, as elicited through prompts that provide task context and auxiliary information. We evaluate our approach on visually-complex, long horizon RL tasks in Minecraft and robot navigation in Habitat. We find that our policies trained on embeddings from off-the-shelf, general-purpose VLMs outperform equivalent policies trained on generic, non-promptable image embeddings. We also find our approach outperforms instruction-following methods and performs comparably to domain-specific embeddings. Finally, we show that our approach can use chain-of-thought prompting to produce representations of common-sense semantic reasoning, improving policy performance in novel scenes by 1.5 times.

## 1 Introduction

Embodied decision-making often requires representations informed by world knowledge for perceptual grounding, planning, and control. Humans rapidly learn to perform sensorimotor tasks by drawing on prior knowledge, which might be high-level and abstract ("If I'm cooking something that needs milk, the milk is probably in the refrigerator") or grounded and low-level (e.g., what refrigerators and milk look like). These capabilities would be highly beneficial for reinforcement learning (RL) too: we aim for our agents to interpret tasks in terms of concepts that can be reasoned about with relevant prior knowledge and grounded with previously-learned representations, thus enabling more efficient learning. However, doing so requires a condensed source of vast amounts of general-purpose world knowledge, captured in a form that allows us to specifically index into and access *task-relevant* information. Therefore, we need representations that are contextual, such that agents can use a concise task context to draw out relevant background knowledge, abstractions, and grounded features that aid it in acquiring a new behavior.

An approach to facilitate this involves integrating RL agents with the prior knowledge and reasoning abilities of pre-trained foundation models. Transformer-based language models (LMs) and vision-language models (VLMs) are trained on Internet-scale data to enable generalization in downstream tasks requiring facts or common sense. Moreover, in-context learning (Brown et al., 2020), chain-of-thought reasoning (CoT) (Wei et al., 2023), and instruction fine-tuning (Ouyang et al., 2022) have provided better ways to index into (V)LMs' knowledge and steer their capabilities based on user needs. These successes have seen some transfer to embodied control, with (V)LMs being used to reason about goals to produce executable plans (Ahn et al., 2022) or as encoders of useful information (like instructions (Liu et al., 2023) or feedback (Sharma et al., 2023)) that the control policy utilizes. Both these paradigms have major limitations: actions generated by LMs are often not appropriately grounded, unless the tasks and scenes are amenable to being expressed or captioned in language. Even then, (V)LMs are often only suited to producing subtask plans, not low-level control signals. On the other hand, using (V)LMs to simply encode inputs under-utilizes their knowledge and reasoning abilities, instead focusing on producing embeddings that reflect the compositionality of language (e.g., so an instruction-following policy may generalize). This motivates the development of an algorithm

Figure 1: **Example instantiations of PR2L for tasks in Minecraft and Habitat.** We query a VLM with a *task-relevant prompt* about observations to produce *promptable representations*, which we train a policy on via RL. Rather than directly asking for actions or specifying the task, the prompt enables indexing into the VLM's prior world knowledge to access task-relevant information. This prompt also allows us to inject auxiliary information and elicit chain-of-thought reasoning.

for learning to produce low-level actions that are grounded and leverage (V)LMs' knowledge and reasoning.

To this end, we introduce **P**romptable **R**epresentations for **R**einforcement **L**earning (**PR2L**): a flexible framework for steering VLMs into producing *semantic features*, which **(i)** integrate observations with prior task knowledge and **(ii)** are grounded into actions via RL (see Figure 1). Specifically, we ask a VLM questions about observations that are related to the given control task, priming it to attend to task-relevant features in the image based on both its internal world knowledge, reasoning capabilities, and any supplemental information injected via prompting. The VLM then encodes this information in decoded text, which is discarded, and associated embeddings, which serve as inputs to a learned policy. In contrast to the standard approach of using pre-trained image encoders to convert visual inputs into *generic* features for downstream learning, our method yields *task-specific* features capturing information particularly conducive to learning a considered task. Thus, the VLM does not just produce an un-grounded encoding of instructions, but embeddings containing semantic information relevant to the task, that is both grounded and informed by the VLM's prior knowledge.

To the best our knowledge, we introduce the first approach for initializing RL policies with generative VLM representations. We demonstrate our approach on tasks in Minecraft (Fan et al., 2022) and Habitat (Savva et al., 2019), as they present semantically-rich problems representative of many practical, realistic, and challenging applications of RL. We find that PR2L outperforms equivalent policies trained on vision-only embeddings or with instruction-conditioning, popular ways of using pre-trained image models and VLMs respectively for control. We also show that promptable representations extracted from general-purpose VLMs are competitive with domain-specific representations. Our results highlight how visually-complex control tasks can benefit from accessing the knowledge captured within VLMs via prompting in both online and offline RL settings.

## 2 RELATED WORKS

**Vision-language models.** In this work, we utilize *generative VLMs* (like Li et al. (2022; 2023a); Dai et al. (2023); Karamcheti et al. (2024)): models that generate language in response to an image and a text prompt passed as input. This is in contrast to other designs of combining vision and language that either generate images or segmentation (Rombach et al., 2022; Kirillov et al., 2023) and contrastive representations (Radford et al., 2021). Formally, the VLM enables sampling from $p(x_{1:K}|I, c)$, where $x_{1:K}$ represents the $K$ tokens of the output, $I$ is the input image(s), $c$ is the prompt, and $p$ is the distribution over natural language responses produced by the VLM on those inputs. Typically, the VLM is pre-trained on tasks that require building association between vision and language such as captioning. All these tasks require learning to attend to certain semantic features of input images depending on the given prompt. For auto-regressive generative VLMs, this distribution is factorized as $\prod_t p(x_t|I, c, x_{1:t-1})$. Typical architectures parameterize these distributions using weights that define a representation $\phi_t(I, c, x_{1:t-1})$, which depends on the image $I$, the prompt $c$, and the previously emitted tokens, and a decoder $p(x_t|\phi_t(I, c, x_{1:t-1}))$, which defines a distribution over the next token.

**Embodied (V)LM reasoning.** Many recent works have leveraged (V)LMs as priors over effective plans for a given goal. These works use the model's language modeling and auto-regressive generation capabilities to extract such priors as textual subtask sequences (Ahn et al., 2022; Huang et al., 2022b; Sharma et al., 2022) or code (Liang et al., 2023; Singh et al., 2022; Zeng et al., 2022; Vemprala

et al., 2023), thereby using the (V)LM to decompose long-horizon tasks into executable parts. These systems often need grounding mechanisms to ensure plan feasibility (e.g., affordance estimators (Ahn et al., 2022), scene captioners (Zeng et al., 2022), or trajectory labelers (Palo et al., 2023)). They also often assume access to low-level policies that can execute these subtasks, such as robot pick-and-place skills (Ahn et al., 2022; Liang et al., 2023), which is often a strong assumption. These methods generally do not address how such policies can be acquired, nor how these low-level skills can themselves benefit from the prior knowledge in (V)LMs. Even works in this area that use RL still use (V)LMs as state-dependent priors over reasonable high-level goals to learn (Du et al., 2023). This is a key difference from our work: instead of considering priors on plans/goals, we rely on VLM's implicit knowledge *of the world* to extract representations which encode task-relevant information. We train a policy to convert these features into low-level actions via standard RL, meaning the VLM does not need to know how to take actions for a task.

**Embodied (V)LM pre-training.** Other works use (V)LMs to embed useful information like instructions (Liu et al., 2023; Myers et al., 2023; Lynch and Sermanet, 2021; Mees et al., 2023; O.M.T. et al., 2023), feedback (Sharma et al., 2023; Bucker et al., 2022), reward specifications (Fan et al., 2022), and data for world modeling (Lin et al., 2023b; Narasimhan et al., 2018). These works use (V)LMs as *encoders* of the compositional semantic structure of input text and images, which aids in generalization: an instruction-conditioned model may never have learned to grasp apples (but can grasp other objects), but by interacting with them in other ways and receiving associated language descriptions, the model might still be able to grasp them zero-shot. In contrast, our method produces embeddings that are informed by world knowledge and reasoning, both from prompting and pre-training. Rather than just specifying that the task is to acquire an apple, we ask a VLM to parse observations into task-relevant features, like whether there is an apple in the image or if the observed location likely contains apples – information that is useful even in single-task RL. Thus, we use VLMs to help RL solve new tasks, not just to follow instructions.

These two categories are not mutually exclusive: Brohan et al. (2023a) use VLMs to understand instructions, but also reasoning (e.g., figuring out the "correct bowl" for a strawberry is one that contains fruits); Palo et al. (2023) use a LM to reason about goal subtasks and a VLM to know when a trajectory matches a subtask, automating the demonstration collection/labeling of Ahn et al. (2022), while Adeniji et al. (2023) use a similar approach to pretrain a language-conditioned RL policy that is transferable to learning other tasks; and Shridhar et al. (2021) use CLIP to merge vision and text instructions directly into a form that a Transporter (Zeng et al., 2020) policy can operationalize. Nevertheless, these works primarily focus on instruction-following for robot manipulation. Our approach instead prompts a VLM to supplement RL with representations of world knowledge, not instructions. In addition, except for Adeniji et al. (2023), these works focus on behavior cloning (BC), assuming access to demonstrations for policy learning, whereas our framework can be used for both online RL and offline RL/BC.

## 3 PR2L: PROMPTABLE REPRESENTATIONS FOR REINFORCEMENT LEARNING

We adopt the standard framework of partially-observed Markov decision process in deep RL, wherein the objective is to find a policy mapping states to actions that maximizes the expected returns. Our goal is to supplement RL with task-relevant information extracted from VLMs containing general-purpose knowledge. One way to index into this information is by prompting the model to get it to produce semantic information relevant to a given control task. Therefore, our approach, PR2L, queries a VLM with a task-relevant prompt for each visual observation received by the agent, and receives both the decoded text and, critically, the intermediate representations, which we refer to as *promptable representations*. Even though the decoded text might often not be correct or directly usable for choosing the action, our key insight is that these VLM embeddings can still provide useful semantic features for training control policies via RL. This recipe enables us to incorporate semantic information without the need of re-training or fine-tuning a VLM to directly output actions, as proposed by Brohan et al. (2023a). Note that our method is *not* an instruction-following method, and it does **not** require a task instruction to perform well. Instead, our approach still learns control via RL, while benefiting from the incorporation of *background context*. In this section, we will describe various components of our approach, accompanied by practical design choices and considerations.

### 3.1 PROMPTABLE REPRESENTATIONS

In principle, one can directly query a VLM to produce actions for a task given a visual observation. While this may work when high-level goals or subtasks are sufficient, VLMs are empirically poor at

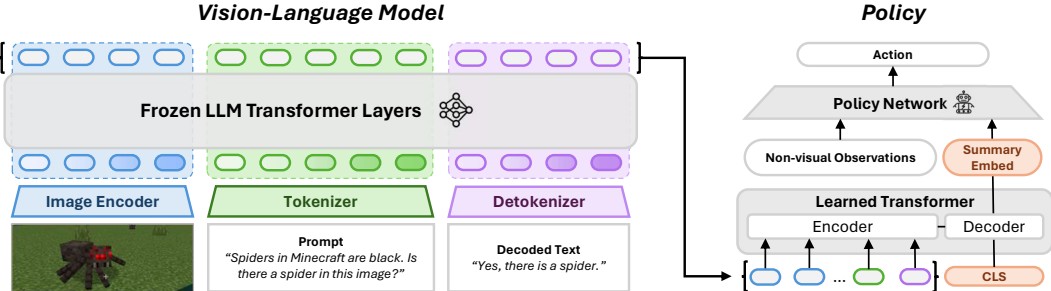

Figure 2: **Schematic of how we extract task-relevant features from the VLM and use them in a policy trained with RL.** These representations can incorporate task context from the prompt, while generic image embeddings cannot. As generative VLM's embeddings can be variable length, the policy has a Transformer layer that takes in these embeddings and a "CLS" token, thereby condensing all inputs into a single summary vector.

yielding the low-level actions used commonly in RL (Huang et al., 2022a). As VLMs are trained to follow instructions and answer questions about images, it is more appropriate to use these models to extract and reason about *semantic features* about observations that are conducive to being linked to actions. We thus elicit features that are useful for the downstream task by querying these VLMs with *task-relevant prompts* that provide contextual task information, thereby causing the VLM to attend to and interpret appropriate parts of observed images. Extracting these features naïvely by only using the VLM's *decoded text* has its own challenges: such models often suffer from hallucinations (Ji et al., 2023) and an inability to report what they "know" in language, even when their embeddings contain such information (Kadavath et al., 2022; Hu and Levy, 2023). However, even when the text is bad, the underlying *representations* still contain valuable granular world information that is potentially lost in the projection to language (Li et al., 2021; Wiedemann et al., 2019; Huang et al., 2023; Li et al., 2023b). Thus, we disregard the generated text and instead provide our policy the embeddings produced by the VLM in response to prompts asking about relevant semantic features in observations instead.

**Which parts of the network can be used as promptable representations?** The VLMs we consider are all based on the Transformer architecture (Vaswani et al., 2017), which treats the prompt, input image(s), and decoded text as token sequences. This architecture provides a source of learned representations by computing embeddings for each token at every layer based on the previous layer's token embeddings. In terms of the generative VLM formalism introduced prior, a Transformer-based VLM's representations $\phi_t(I, c, x_{1:t-1})$ consist of $N$ embeddings per token (the outputs of the $N$ self-attention layers) in the input image $I$, prompt $c$, and decoded text $x_{1:t-1}$. The decoder $p(x_t|\phi_t)$ extracts the final layer's embedding of the most recent token $x_{t-1}$, projecting it to a distribution over the token vocabulary and allowing for it to be sampled. When given a visual observation and task prompt, the tokens representing the prompt, image, and answer consequently encode task-relevant semantic information. Thus, for each observation, we use the VLM to sample a response to the task prompt $x_{1:K} \sim p(x_{1:K}|I, c)$. We then use some or all of these token embeddings $\phi_K(I, c, x_{1:t-1})$ as our promptable representations and feed them, along with any non-visual observation information, as a state representation into our neural policy trained with RL.

In summary, our approach involves creating a task-relevant prompt that provides context and auxiliary information. This prompt, alongside the current visual observation from the environment, is fed to into the VLM to generate tokens. While these tokens are used for decoding, they are ultimately discarded. Instead, we utilize the *representations* produced by the VLM (associated with the image, prompt, and decoded text) as input for our policy, which is trained via an off-the-shelf online RL algorithm to produce appropriate actions. A schematic of our approach is depicted in Figure 2 and a code snippet example is presented in Appendix I.

## 3.2 DESIGN CHOICES FOR PR2L

To instantiate this idea, we need to make some concrete design choices in practice. First, the representations of the VLM's decoded text depend on the chosen decoding scheme: greedy decoding is fast and deterministic, but may yield low-probability decoded tokens; beam search improves on this by considering multiple "branches" of decoded text, at the cost of requiring more compute time (for potentially small improvements); lastly, sampling-based decoding can quickly yield estimates of the

maximum likelihood answer, but at the cost of introducing stochasticity, which may increase variance. Given the inherent high-variance of our tasks (due to sparse rewards and partial observability) and the expense of VLM decoding, we opt for greedy decoding or fixed-seed sampling.

Second, one must choose which VLM layers' embeddings to utilize in the policy. While theoretically, all layers of the VLM could be used, pre-trained Transformer models tend to encode valuable high-level semantic information in their later layers (Tenney et al., 2019; Jawahar et al., 2019). Thus, we opt to only feed the final few layers' representations into our policy. As these representation sequences are of variable length, we incorporate an encoder-decoder Transformer layer in the policy. At each time step in a trajectory, this layer receives variable-length VLM representations, which are attended to and converted into a fixed-length summarization by the embeddings of a learned "CLS" token (Devlin et al., 2019) in the decoder (green in Figure 2). We also note that this policy can receive the observed image directly (e.g., after being embedded by the image encoder), so as to not lose any visual information from being processed by the VLM. However, we do not do this in our experiments in order to more clearly isolate and demonstrate the usefulness of the VLM's representations in particular.

Finally, while it is possible to fine-tune the VLM for RL end-to-end with the policy (Brohan et al., 2023a),this incurs substantial compute, memory, and time overhead, particularly with larger VLMs. Nonetheless, we find that our approach performs better than not using the language and prompting components of the VLM. This holds true even when the VLM is frozen, and only the policy is trained via RL, or when the decoded text occasionally fails to answer the task-specific prompt correctly.

## 3.3 TASK-RELEVANT PROMPT DESIGN

**How do we design good prompts to elicit useful representations from VLMs?** As we aim to extract good state representations from the VLM for a downstream policy, we do not use instructions or task descriptions, but task-relevant prompts: questions that make the VLM attend to and encode semantic features in the image that are useful for the RL policy learning to solve the task (Borja-Diaz et al., 2022). For instance, if the task is to find a toilet within a house, appropriate prompts include "What room is this?" and "Would a toilet be found here?" Intuitively, the answers to these questions help determine good actions (e.g., look around the room or explore elsewhere), making the corresponding representations good for representing the state for a policy. Answering the questions will require the VLM to attend to task-relevant features in the scene, relying on the model's internal conception of what things look like and common-sense semantic relations. One can also prompt the VLM to use chain of thought (Wei et al., 2023) to explain its generated text, often requiring it to reason about task-relevant features in the image, resulting in further enrichment of the state representations. Finally, prompts can provide helpful auxiliary information: e.g., one can describe what certain entities of interest look like, aiding the VLM in detecting them even if they were not commonly found in the model's pre-training data.

Note that prompts based on instructions or task descriptions do not enjoy the above properties: while the goal of those prior methods is to be able to directly query the VLM for the optimal action, the goal of task-relevant prompts is to produce a useful state representation, such that running RL with them can accelerate learning an optimal policy. While the former is not possible without task-specific training data for the VLM in the control task, the latter proves beneficial with off-the-shelf VLMs.

**Evaluating and designing prompts for RL.** Since the specific representations elicited from the VLM are determined by the prompt, we want to design prompts that produce promptable representations that maximize performance on the downstream task. The brute-force approach would involve running RL with each candidate prompt to measure its efficacy, but this would be computationally very expensive. In lieu of this, we evaluate candidate prompts on a small dataset of observations labeled with semantic features of interest for the considered task. Example features include whether task-relevant entities are in the image, the relative position of said entities, or even actions (if expert demonstrations are available). We test prompts by querying the VLM and checking how well the resulting decoded text for each image matches ground truth labels. As this is only practical for small, discrete spaces that are easily expressed in words, we see how well a small model can fit the VLM's embeddings to the labels (akin to probing in self-supervised learning (Shi et al., 2016; Belinkov and Glass, 2019)). While this does not directly optimize for task performance, it does act as a proxy that ensures a prompt's resulting representations encode certain semantic features which are helpful for the task.

## 4 EXPERIMENTAL SETUPS

Our experiments analyze whether promptable representations from VLMs provide benefits to downstream control, thus providing an effective vehicle for transferring Internet-scale knowledge to RL. We aim to show that PR2L is a good source of state representations, even with our current VLMs that are bad at reasoning about actions – as such models become more performant, we expect such representations to be even better. We thus design experiments to answer the following: **(1)** Can promptable representations obtained via task-specific prompts enable more performant and sample-efficient learning than those of non-promptable image encoders pre-trained for vision or control? **(2)** How does PR2L compare to approaches that directly "ask" the VLM to generate good actions for a task specified in the prompt? **(3)** How does PR2L fare against other popular learning approaches or purely visual features in our domains of interest?

### 4.1 DOMAIN 1: MINECRAFT

We first conduct experiments in Minecraft, which provides control tasks that require associating visual observations with rich semantic information to succeed. Moreover, since these observations are distinct from the images in the the pre-training dataset of the VLM, succeeding on these tasks relies crucially on the efficacy of the task-specific prompt in meaningfully affecting the learned representation, enabling us to stress-test our method. E.g., while spiders in Minecraft somewhat resemble real-life spiders, they exhibit stylistic exaggerations such as bright red eyes and a large black body. If the task-specific prompt is indeed effective in informing the VLM of these facts, it would produce a representation that is more conducive to policy learning and this would be reflected in task performance. For this domain, we use the half-precision Vicuna-7B version of the InstructBLIP instruction-tuned generative VLM (Dai et al., 2023; Chiang et al., 2023) to produce promptable representations.

**Minecraft tasks.** We consider all programmatic Minecraft tasks evaluated by Fan et al. (2022): *combat spider*, *milk cow*, *shear sheep*, *combat zombie*, *combat enderman*, and *combat pigman*[1]. The remaining tasks considered by Fan et al. (2022) are creative tasks, which do not have programmatic reward functions or success detectors, so we cannot directly train RL agents on them. We follow the MineDojo definitions of observation/action spaces and reward function structures for these tasks: at each time step, the policy observes an egocentric RGB image, its pose, and its previously action; the policy can choose a discrete action to turn the agent by changing the agent's pitch and/or yaw in discrete increments, move, attack, or use a held item. These tasks are long horizon, with a maximum episode length of 500 - 1000 and taking roughly 200 steps for a learned policy to complete them. See Figure 3 for example observations and Appendix B.1 for more details.

**Comparisons.** We compare PR2L to five performant classes of approaches for RL in Minecraft: **(a)** Methods using non-promptable representations of visual observations. This does not use prompting altogether, instead using task-agnostic embeddings from the VLM's image encoder (specifically, the ViT-g/14 from InstructBLIP – blue in Figure 2). While these representations are still pre-trained, PR2L utilizes prompting to produce *task-specific* representations. For a fair comparison, we use the *exact same* policy architecture and hyperparameters for this baseline as in PR2L, ensuring that performance differences come from prompting for better representations from the VLM. **(b)** Methods that directly "asks" the VLM to output actions to execute on the agent. This adapts the approach of Brohan et al. (2023a) to our setting and directly outputs the action from the VLM. While Brohan et al. (2023a) also fine-tune the VLM backbone, we are unable to do so using our compute resources. To compensate, we do not just execute the action from the VLM, but train an RL policy to map this decoded action to a better one. Note that if the VLM already decodes good action texts, simply copying over this action via RL should be easy. **(c)** Methods for efficient RL from pixels via model-based approaches. We choose Dreamer v3, since it has proven to be successful at learning Minecraft tasks from scratch Hafner et al. (2023). **(d)** Methods leveraging pretrained representations specifically useful for embodied control, though which are non-promptable and non-Minecraft specific. We choose VC-1 and R3M Majumdar et al. (2023); Nair et al. (2022). **(e)** Methods using models pretrained on large-scale Minecraft data. These serve as "oracle" comparisons, as these representations are explicitly fine-tuned on Minecraft YouTube videos, whereas our pre-trained VLM is both frozen and not trained on any Minecraft video data. We choose MineCLIP, VPT, and STEVE-1 as our

---

[1] Fan et al. (2022) also consider *hunt cow/sheep*. However, we omit them as we were unable to replicate their results on those tasks; all approaches failed to learn them.

| | PR2L Prompt | RT-2-style Baseline Prompt | Change Auxiliary Text Ablation Prompt |
|---|---|---|---|
| *Combat Spider* | Spiders in Minecraft are black. Is there a spider in this image? | I want to fight a spider. I can attack, move, or turn. What should I do? | Is there a spider in this image? |
| *Milk Cow* | Is there a cow in this image? | I want to milk a cow. I can use my bucket, move, or turn. What should I do? | Cows in Minecraft are black and white. Is there a cow in this image? |
| *Shear Sheep* | Is there a sheep in this image? | I want to shear a sheep. I can use my shears, move, or turn. What should I do? | Sheep in Minecraft are usually white. Is there a sheep in this image? |
| *Other Combat Tasks* | Is there a [target entity] in this image? | I want to fight a [target entity]. I can attack, move, or turn. What should I do? | - |

Table 1: Prompts used in Minecraft for querying the VLM with PR2L, comparison (b), and the change auxiliary text ablation. For the last column, we remove the auxiliary text for *combat spider*, and add it in for the other two.

sources of Minecraft-specific representations Fan et al. (2022); Baker et al. (2022); Lifshitz et al. (2023).

We use PPO (Schulman et al., 2017) as our base RL algorithm for all non-Dreamer Minecraft policies. We also note that we do *not* compare against non-RL methods, such as Voyager (which uses LLMs to write high-level code skills, abstracting away low-level control to hand-written APIs that use oracle information). See Appendix B.2 for training details and E.1 for further discussion of such non-learned systems.

### 4.2 DOMAIN 2: HABITAT

A major advantage of VLMs pre-trained on Internet-scale data is their reasoning and generalization capabilities. To evaluate this, we run offline BC and RL experiments in the Habitat household simulator. In contrast to Minecraft, tasks in this domain require connecting *naturalistic* images with real-world common sense about the structure and contents of typical home environments. Our experiments evaluate (**1**) whether PR2L confers the generalization properties of VLMs to our policies, (**2**) whether PR2L-based policies can leverage the semantic reasoning capabilities of the underlying VLM (e.g., via chain-of-thought Wei et al. (2023)), and (**3**) whether PR2L can learn entirely from stale, offline data sources. We use a Llama2-7B Prismatic VLM for the Habitat experiments Karamcheti et al. (2024).

**Habitat tasks.** We consider the ObjectNav task suite in 3D scanned household scenes from the HM3D dataset (Savva et al., 2019; Yadav et al., 2023a; Ramakrishnan et al., 2021). These tasks involve a simulated robot traversing a home environment to find an instance of a specified object (toilet, bed, sofa, television, plant, or chair) in the shortest path possible. The full benchmark consists of 80 household scenes intended to train the agent and 20 for validation. We change the observation space to consist of just RGB vision, previous action, pose, and target object class, omitting depth images to ensure that observed performance differences come from the quality of promptable representations vs. unpromptable ones. Like with MineDojo, these tasks are long horizon, taking 80 steps for a privileged shortest path follower to succeed and 150+ for humans. See Figure 3 for example observations and Appendix C for more details.

**Comparisons.** To see if PR2L can leverage VLM reasoning capabilities, we train two PR2L policies, one with and one without chain-of-thought prompting (see Section 4.3). We also train a policy on Prismatic VLM image encoder embeddings (equivalent to Minecraft approach (a), but with Dino+SigLIP Caron et al. (2021); Zhai et al. (2023)) on a human demonstration dataset collected from the ObjectNav training scenes collected with Habitat-Web Ramrakhya et al. (2022) and used by past works on large-scale BC on pre-trained visual representations Ramrakhya et al. (2023); Yadav et al. (2023b); Majumdar et al. (2023). As it previously achieved state-of-the-art performance among those works, we also compare against two policies using VC-1 as an encoder (Majumdar et al., 2023), either using just its summarizing CLS token or using a learned Transformer layer to condense its patch embeddings. We adopt the same LSTM-based recurrent architecture used by that work, but replace the image embeddings with a learned Transformer layer that condenses our input token embeddings (from the VLM, VLM image encoder, or VC-1) into a single summary embedding, as done with Minecraft.

Due to computational constraints, we train all policies on just under a tenth of the full dataset of 77k trajectories/12M steps. In contrast, other works using this dataset train on the entire dataset. Nevertheless, we evaluate on the unseen validation scenes, thereby testing how well PR2L generalizes.

### 4.3 DESIGNING TASK-SPECIFIC PROMPTS FOR MINECRAFT AND HABITAT

We now discuss how to design prompts for PR2L. As noted in Section 3.3, these are not instructions or task descriptions, but prompts that force the VLM to encode semantic information useful for the

| Task | PR2L (Ours) | Baselines | | | | | Oracles | | |
|---|---|---|---|---|---|---|---|---|---|
| | | VLM Image Encoder | RT-2-style | Dreamer | VC-1 | R3M | MineCLIP | VPT | STEVE-1 |
| *Combat Spider* | **97.6 ± 14.9** | 51.2 ± 9.3 | 71.5 ± 9.7 | 5.4 ± 1.1 | 72.2 ± 9.3 | 72.9 ± 8.7 | *176.9 ± 19.8* | *137.2 ± 19.2* | 88.8 ± 14.0 |
| *Milk Cow* | **223.4 ± 35.4** | 95.2 ± 18.7 | 128.6 ± 28.9 | 24.0 ± 1.2 | 96.6 ± 16.3 | 100.0 ± 14.1 | 194.4 ± 33.3 | 85.5 ± 14.5 | 75.2 ± 15.4 |
| *Shear Sheep* | **37.0 ± 4.4** | 23.0 ± 3.6 | 26.2 ± 3.2 | 20.9 ± 1.2 | 26.5 ± 4.0 | 17.5 ± 2.4 | 23.1 ± 3.7 | 24.1 ± 2.9 | 18.2 ± 2.5 |
| *Combat Zombie* | **24.6 ± 1.6** | 14.8 ± 2.0 | 18.2 ± 2.1 | 1.8 ± 0.2 | 5.6 ± 1.0 | 5.8 ± 1.4 | *56.6 ± 8.3* | *31.2 ± 3.2* | 23.6 ± 3.4 |
| *Combat Enderman* | **52.2 ± 5.6** | 51.9 ± 6.8 | 44.6 ± 5.8 | 1.6 ± 0.5 | 27.2 ± 2.4 | 33.8 ± 3.8 | *72.1 ± 7.1* | *74.4 ± 13.2* | *59.3 ± 6.7* |
| *Combat Pigman* | **46.4 ± 3.3** | 36.8 ± 3.7 | 35.1 ± 2.5 | 5.8 ± 1.5 | 33.7 ± 4.9 | 31.4 ± 4.2 | *189.0 ± 7.9* | *169.0 ± 7.8* | *98.3 ± 8.4* |

Table 2: **Performance of PR2L, baseline, and oracle approaches in Minecraft tasks.** Values reported are IQM successes and standard errors. PR2L universally outperforms all baselines. As they are trained on Minecraft-specific data, the oracles outperform PR2L in half the comparisons (italicized).

| Task (# Episodes) | PR2L (Ours) | | VLM Image Encoder | VC-1 + CLS | | VC-1 + Patch Embeds | |
|---|---|---|---|---|---|---|---|
| | With CoT | Without CoT | | 40 Epochs | 120 Epochs | 40 Epochs | 120 Epochs |
| *Average (2000)* | **41.9%** | 27.8% | 11.6% | 6.8% | 8.9% | 13.6% | 15.8% |
| *Toilet (398)* | **37.2%** | 22.9% | 8.8% | 2.8% | 2.0% | 7.0% | 9.3% |
| *Bed (433)* | **45.0%** | 28.9% | 12.9% | 6.7% | 9.9% | 14.8% | 19.2% |
| *Sofa (376)* | **48.1%** | 34.3% | 11.7% | 9.8% | 14.4% | 17.0% | 19.4% |
| *Chair (428)* | **51.2%** | 40.9% | 17.5% | 11.7% | 15.0% | 22.4% | 23.8% |
| *Television (281)* | **26.7%** | 10.3% | 5.0% | 2.8% | 3.2% | 4.6% | 4.6% |
| *Plant (84)* | **23.8%** | 8.3% | 9.1% | 1.2% | 1.2% | 9.5% | 9.5% |

Table 3: **Performance of PR2L and baselines on Habitat ObjectNav tasks.** Following prior works, values reported are average success rates in unseen validation scenes. PR2L (with or without CoT) does better than all other approaches. PR2L with CoT does the best, universally achieving more than double the performance of all non-PR2L approaches and 14.7% higher average performance than PR2L without CoT. Note that PR2L and image encoder policies were trained for 40 epochs, but VC-1 policies' performance saturated at 120, so we report their performance at both times.

task in its representation. The simplest relevant feature for our Minecraft tasks is the presence of the target entity in an observation. Thus, we choose "Is there a [target entity] in this image?" as the base of our chosen prompt. We also pick two alternate prompts per task that prepend different amounts of auxiliary information about the target entity. E.g., for *combat spider*, one candidate is "Spiders in Minecraft are black." To choose between these candidates, we measure how well the VLM is able to decode a correct answer to the prompt question of whether or not the target entity is present in the image on a small annotated dataset. Full details of this prompt evaluation scheme for the first three Minecraft tasks are presented in Appendix A and Table 5. We find that auxiliary text only helps with detecting spiders while systematically and significantly degrading the detection of sheep and cows. Our ablations show that this detection success rate metric correlates with performance of the RL policy. Additionally, the prompts used for comparison (b) follow the prompt structure prescribed by Brohan et al. (2023a), which motivated this comparison. In these prompts, we also provide a list of actions that the VLM can choose from to the policy. All chosen prompts are presented in Table 1.

For Habitat, we choose the prompt "Would a [target object] be found here? Why or why not?" As opposed to the Minecraft prompts, this does not just identify the presence of a target object in the image, but draws on general knowledge from the VLM to determine if the observed location would contain the target object, even if said object is not in view. The second part of the prompt then leads the VLM to provide a chain of thought (CoT) (Wei et al., 2023) rationale for its final answer. This CoT draws out task-relevant VLM world knowledge by explicitly reasoning about visual semantic concepts, that are useful to learning a policy (see Table 4). To investigate if PR2L enables embodied agents to benefit from these VLM common-sense reasoning capabilities (even if they do not directly reason about actions), we train PR2L policies both with and without the second part of the prompt.

## 5 RESULTS

**Minecraft results.** We report the interquartile mean (IQM) and standard error number of successes over 16 seeds for all Minecraft tasks in Table 2. PR2L uniformly outperforms the non-oracle approaches of (a) using non-promptable image embeddings, (b) directly asking the VLM for actions, (c) learning from scratch Dreamer, and (d) using non-promptable control-specific embeddings.

PR2L outperforms **(a) the VLM image encoder baseline**, even though both approaches receive the same visual features, with PR2L simply transforming those features via prompting an LLM (with no additional information from the environment), thus supporting that prompting does shape representations in a beneficial way for learning control tasks. We provide an analysis of why PR2L states are better than **(b) RT-2-style** ones in Appendix H.1. We observe that PR2L embeddings are bimodally distributed, with transitions leading to high reward clustered at one mode. This structure likely enables more efficient learning, thereby showing how control tasks can benefit from extracting

prior knowledge encoded in VLMs by prompting them with task context, even when the VLM does not know how to act. For **(c) the model-based comparisons**, we find that Dreamer is not as conducive at learning our Minecraft tasks. We hypothesize this is because our tasks are comparatively shorter than the ones considered by Hafner et al. (2023), so learning a model is less beneficial (while PR2L provides immediately-useful representations). Additionally, we note that all our approaches involve interacting with partially-observable, non-stationary entities, which the Dreamer model may have a hard time learning. See Appendix E.2 for further discussion. Finally, **(e) the oracles** outperform PR2L in *combat enderman/pigman*, all but STEVE-1 do better in *combat spider/zombie*, and none do better in *shear sheep/milk cow*. We hypothesize this is because endermen and pigmen are Minecraft-specific entities, giving rise to comparatively poor representations in the VLM (which is trained exclusively on natural images). In contrast, Minecraft zombies/spiders are heavily stylized, but still somewhat resemble other depictions of such creatures, while Minecraft cows and sheep are the closest to their naturalistic counterparts, making PR2L more effective. Even though our VLM is not trained on Minecraft data, its representations yield better policies in half the oracle comparisons.

We provide ablations in Table 8 and Appendix F. We find that (1) PR2L performs worse when it is unprompted or does not decode text, (2) our prompt evaluation scheme successfully identified cases where auxiliary text improves/degrades performance, and (3) a policy with oracle entity detection does worse than PR2L, suggesting our prompt is not just eliciting that feature from the VLM.

**Habitat results.** Following prior works, we report success rates on the ObjectNav validation episodes in Table 3. PR2L with CoT outperforms all other policies on all tasks, including an almost $4\times$ performance increase over the VLM image encoder baselines – again, suggesting that using promptable representations for control improves over the base purely-visual embeddings. While PR2L without CoT still does better than all baselines, we find CoT prompting improves policy performance (by $1.5\times$, from 27.8% success rate to 41.9%), likely because it provides the policy with useful generalizable features: e.g., even if the agent comes across an unfamiliar room while searching for a toilet, it still knows to look elsewhere if the VLM reasons that, due to the presence of a bed, the room is likely a bedroom (which is unlikely to contain toilets). Thus, even if the VLM cannot reason about actions, *our results indicate that PR2L provides a promising way of using its ability to reason about image semantics and common sense for control*. See Table 4 for CoT examples.

While we do not beat VC-1's reported SOTA BC performance (60.3% success rate when VC-1 is frozen Majumdar et al. (2023)), we note that said performance is achieved with (1) over ten times more training data and gradient steps and (2) image augmentations to prevent overfitting. Our VC-1 policies were trained on the same amount of data as our PR2L agent and for $1\text{-}3\times$ as many gradient steps, but perform far worse, suggesting that PR2L is significantly more sample- and compute-efficient than VC-1 policies. Additionally, PR2L does not use any explicit countermeasures to overfitting, yet still generalizes well to unseen ObjectNav scenes (aided by the VLM's representations of reasoning).

Finally, we analyze policies trained with offline RL in a simplified Habitat setting in Appendices D, H, where we find that VLM representations align well with the returns of an optimal policy.

| Target | Example VLM Responses to "Would a [target] be found here? Why or why not?" |
|---|---|
| *Toilet* | "No, a toilet is not found in this room as it is a bedroom. Toilets are typically found in bathrooms . The image shows a large bed with a wooden headboard, nightstands, and lamps ." |
| | "Yes, a toilet would be found in this room because it is a bathroom. The presence of a toilet in the image is expected, as bathrooms are designed to accommodate toilets for personal hygiene ." |
| *Bed* | "No, a bed would not be found in this room because it is a dining room. The presence of a dining table and chairs suggests that this room is designed for eating meals and socializing, rather than sleeping " |
| | "Yes, a bed would be found here because the room is described as a bedroom. It has a bed , a television, a lamp, and a dresser , all of which are common furnishings in a bedroom ." |
| *Sofa* | "No, a sofa is not found here because it is a kitchen area with appliances and cabinets . Sofas are typically found in living rooms or other common areas ." |
| | " Yes, there is a black leather sofa in the living room. It has a red pillow on it. It is a large sectional couch ." |

Table 4: Example VLM responses to the Habitat prompt for various images. Beyond just detecting the target, prompting the VLM for CoT elicits relevant common sense, which it semantically relates to other useful visual features. By using the underlying VLM embeddings as a state representation, the policy thus integrates the VLM's knowledge and reasoning into its decision-making.

## 6 CONCLUSION

We propose Promptable Representations for Reinforcement Learning, a method for extracting semantic features from images by prompting VLMs with task context to leverage their extensive general-purpose prior knowledge. We demonstrate PR2L in Minecraft and Habitat, domains that benefit from interpreting observations in terms of semantic concepts that can be related to task context. This framework for using VLMs for control opens new directions. For example, other types of foundation models pre-trained with more sophisticated methods could also be used for PR2L: e.g., ones trained on physical interactions might yield features which encode physics or action knowledge, rather than just common-sense visual semantics. Developing and using such models with PR2L offers an exciting way to transfer diverse prior knowledge to a broad range of control applications.

A limitation of PR2L is that prompts are currently hand-crafted based on the user's conception of useful task features. While coming up with good prompts for our tasks was not hard, the process of evaluating and improving them could be automated, which we leave to future works. We also find that the quality of representations largely depends on the VLM – e.g., InstructBLIP could not reason well about Habitat scenes, but the more recent Prismatic VLMs are more capable in that regard, enabling our CoT experiments. Thus, as VLM capabilities are expected to increase, we expect the quality of their representations to also improve. Lastly, the size and speed of VLMs can limit their applicability. Our policies typically achieve 3-5 Hz inference speeds, comparable to those of robot policies built on large models Brohan et al. (2023b;a); O.M.T. et al. (2023). Likewise, our VLM sizes are comparable to models used for policies in prior works (Brohan et al., 2023a; Szot et al., 2024). While their inference speeds may hinder online policy learning, we find that offline approaches (which can parallelize training and data generation) we used for Habitat help remedy this.

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

| Target Entity | Prompt | True Positive Rate | True Negative Rate |
|---|---|---|---|
| Spider | "Is there a spider in this image?" | 22.27% | 100.00% |
| | "Spiders in Minecraft are black. Is there a spider in this image?" | 73.42% | 94.54% |
| | "Spiders in Minecraft are black and have red eyes and long, thin legs. Is there a spider in this image?" | 50.50% | 99.85% |
| Cow | "Is there a cow in this image?" | 71.00% | 45.41% |
| | "Cows in Minecraft are black and white. Is there a cow in this image?" | 98.22% | 2.00% |
| | "Cows in Minecraft are black and white and have four legs. Is there a cow in this image?" | 96.67% | 7.35% |
| Sheep | "Is there a sheep in this image?" | 88.00% | 59.83% |
| | "Sheep in Minecraft are white. Is there a sheep in this image?" | 100.00% | 0.00% |
| | "Sheep in Minecraft are white and have four legs. Is there a sheep in this image?" | 100.00% | 0.00% |

Table 5: InstructBLIP's performance at decoding text indicating that it detected the presence of a target entity when given different prompts. We use this as a proxy metric for prompt engineering for RL, allowing us to determine which prompt to use for PR2L.

## A  PROMPT EVALUATION FOR RL IN MINECRAFT

We discuss how to evaluate prompts to use with PR2L, by showcasing an example for a Minecraft task. We start by noting that the presence and relative location of the entity of interest for each task (i.e., spiders, sheep, or cows) are good features for the policy to have. To evaluate if a prompt elicits these features from the VLM, we collect a small dataset of videos in which each Minecraft entity of interest is on the left, right, middle, or not on screen for the entirety of the clip. Each video is collected by a human player screen recording visual observations from Minecraft of the entity from different angles for around 30 seconds at 30 frames per second (with the exception of the video where the entity is not present, which is a minute long).

We propose prompts that target each of the two features we labeled. First, we evaluate prompts that ask "Is there a(n) [entity] in this image?" As the answers to these questions are just yes/no, we see how well the VLM can directly generate the correct answer for each frame in the collected videos. The VLM should answer "yes" for frames in the three videos where the target entity is on the left, right, or middle of the screen and "no" for the final video. Second, we evaluate if our prompts can extract the entity's relative position (left, right, or middle) in the videos where it is present. We note that the prompts we tried could not extract this feature in the decoded text (e.g., asking "Is the [entity] on the left, right, or middle of the screen?" will always cause the VLM to decode the same text). Thus, we try to see if this feature can be extracted from the decoded texts' representations. We measure this by fitting a three-category linear classifier of the entity's position given the *token-wise mean* of the decoded tokens' final embeddings. This is an unsophisticated and unexpressive classifier, i.e., we do not have to worry about the model potentially memorizing the data, which means that good classification performance corresponds to an easy extractability of said feature.

We evaluate three types of prompts per task entity for the first feature: one simply asking if the entity is present in the image (e.g., "Is there a spider in this image?") and two others adding varying amounts of auxiliary information about visual characteristics of the entity (e.g., "Spiders in Minecraft are black. Is there a spider in this image?" and "Spiders in Minecraft are black and have red eyes and long, thin legs. Is there a spider in this image?"). We present evaluations of all such prompts in Table 5. We find that the VLM benefits greatly from auxiliary information for the spider case only, likely because spiders in Minecraft are the most dissimilar to the ones present in natural images of real spiders, whereas cows and sheep are still comparatively similar, especially in terms of scale and color. However, adding too much auxiliary information degrades performance, perhaps because the input prompt becomes too long, and therefore is out-of-distribution for the types of prompts that the VLM was pre-trained on. This same argument may explain why auxiliary information degrades

**Tasks in MineDojo**

*Combat Spider*

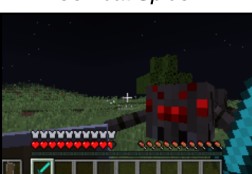

*Milk Cow*

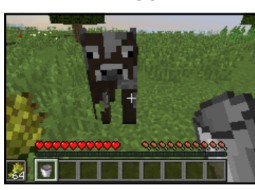

*Shear Sheep*

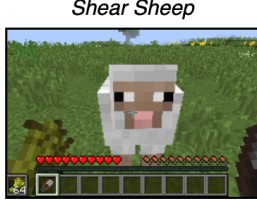

**Tasks in Habitat**

*ObjectNav: Find [toilet / sofa / bed].*

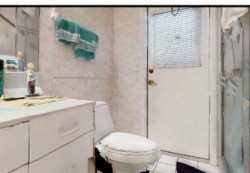

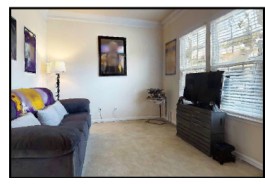

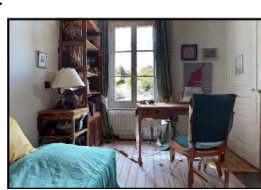

Figure 3: Example tasks, observations, and task-relevant prompts from MineDojo and Habitat.

performance for the other two target entities as well, causing them to almost always answer that said entities are present, even when they are not. Once more, considering that these targets exhibit a higher degree of visual resemblance to to their real counterparts compared to Minecraft spiders, it is reasonable to infer that the VLM would not benefit from auxiliary information. Furthermore, taking into account that the auxiliary information we gave is more common-sense than the information given for the spider, it could imply that the prompts are also more likely to be out-of-distribution (given that "sheep are white" is so obvious that people would not bother expressing it in language), causing the systematic performance degradation.

For the probing evaluation, we find that all three prompts reach similar final linear classifiabilities for each of their target entities, as shown in Figure 4. While this does not aid in choosing one prompt over another, it does confirm that the VLM's decoded embeddings for each prompt still contains this valuable and granular position information about the target entity, *even though the input prompt did not ask for it*.

## B MINEDOJO DETAILS

### B.1 ENVIRONMENT DETAILS

**Spaces.** The observation space for the Minecraft tasks consists of the following:

1. **RGB:** Egocentric RGB images from the agent. (160, 256, 3)-size tensor of integers $\in \{0, 1, ..., 255\}$.

2. **Position:** Cartesian coordinates of agent in world frame. 3-element vector of floats.

3. **Pitch, Yaw:** Orientation of agent in world frame in degrees. Note that we limit the pitch to $15°$ above the horizon to $75°$ below for *combat spider*, which makes learning easier (as the agent otherwise often spends a significant amount of time looking straight up or down). Two 1-element vectors of floats.

4. **Previous Action:** The previous action taken by the agent. Set to no operation at the start of each episode. One-hot vector of size $|\mathcal{A}| = 53$ for *combat spider* and 89 otherwise (see below).

This differs from the simplified observation space used in Fan et al. (2022) in that we do not use any nearby voxel label information and impose pitch limits for *combat spider*. This observation space is the same for all Minecraft experiments.

The action space is discrete, consisting of 53 or 89 different actions:

1. **Turn:** Change the yaw and pitch of the agent. The yaw and pitch can be changed up to $\pm 90°$ in multiples of $15°$. As they can both be changed at the same time, there are $9 \times 9 = 81$ total different turning actions. The turning action where the yaw and pitch changes are both $0°$ is the no operation action. Note that, since we impose pitch limits for the spider task, we also limit the change in pitch to $\pm 30°$, meaning there are only 45 turning actions in that case.

2. **Move:** Move forward, backward, left, right, jump up, or jump forward for 6 actions total.

3. **Attack:** Swing the held item at whatever is targeted at the center of the agent's view.

4. **Use Item:** Use the held item on whatever is targeted at the center of the agent's view. This is used to milk cows or shear sheep (with an empty bucket or shears respectively). If holding a sword and shield, this action will block attacks with the latter.

This non-*combat spider* action space is the same as the simplified one in Fan et al. (2022). All experiments for a given task share the same action space.

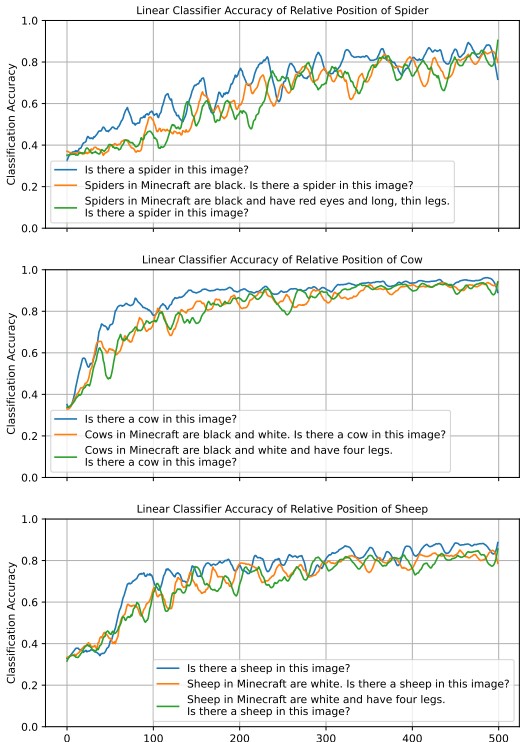

Figure 4: We train a linear classifier to predict the relative position of the target entity (left/right/middle) based on the average VLM embeddings decoded in response to each associated candidate prompt. We find that all three candidate prompts per task elicit embeddings that are similarly highly conducive to this classification scheme.

**World specifications.** MineDojo implements a fast reset functionality that we use. Instead of generating an entirely new world for each episode, fast reset simply respawns the player and all specified entities in the same world instance, but with fully restored items, health points, and other relevant task quantities. This lowers the time overhead of resets significantly, but also means that some changes to the world (like block destruction) are persistent. However, as breaking blocks generally takes multiple time steps of taking the same action (and does not directly lead to any reward), the agent empirically does not break many blocks aside from tall grass (which is destroyed with a single strike from any held item). We keep all reset parameters (like the agent respawn radius, how far away entities can spawn from the agent, etc) at their default values provided by MineDojo.

We stage all tasks in the same area of the same programmatically-generated world: namely, a sunflower plains biome in the world with seed 123. This is the default location for the implementation of the spider combat task in MineDojo. We choose this specific world/location as it represents a prototypical Minecraft scene with relatively easily-traversable terrain (thus making learning faster and easier).

**Additional task details and reward functions.** We provide additional notes about our Minecraft tasks.

*Combat spider*: Upon detecting the agent, the spider approaches and attacks; if the agent's health is depleted, then the episode terminates in failure. The agent receives $+1$ reward for striking any entity and $+10$ for defeating the spider. We also include several distractor animals (a cow, pig, chicken, and

| Hyperparameter | Task | | | | | |
| --- | --- | --- | --- | --- | --- | --- |
| | *Combat Spider* | *Milk Cow* | *Shear Sheep* | *Combat Zombie* | *Combat Enderman* | *Combat Pigman* |
| Total Train Steps | 150000 | | | 100000 | | |
| Rollout Steps | | | | 2048 | | |
| Action Entropy Coefficient | | | | 5e-3 | | |
| Value Function Coefficient | | | | 0.5 | | |
| Max LR | 5e-5 | 1e-4 | 1e-4 | 5e-5 | 1e-4 | 5e-5 |
| Min LR | 5e-6 | 1e-4 | 1e-4 | 5e-6 | 1e-4 | 5e-6 |
| Batch Size | | | | 64 | | |
| Update Epochs | | | | 10 | | |
| $\gamma$ | | | | 0.99 | | |
| GAE $\lambda$ | | | | 0.95 | | |
| Clip Range | | | | 0.2 | | |
| Max Gradient Norm | | | | 0.5 | | |
| Normalize Advantage | | | | True | | |

Table 6: PPO hyperparameters for Minecraft tasks, shared by the baselines, our method, and ablations.

sheep) that passively wander the task space; the agent can reward game by striking these animals, making credit assignment of success rewards and the overall task harder.

*Milk cow*: The agent also holds wheat in its off hand, which causes the cow to approach the agent when detected and sufficiently nearby. For each episode, we track the minimum visually-observed distance between the agent and the cow at each time step. The agent receives $+0.1|\Delta d_{\min}|$ reward for decreasing this minimum distance (where $\Delta d_{\min} \leq 0$ is the change in this minimum distance at a given time step) and $+10$ for successfully milking the cow.

*Shear sheep*: As with *milk cow*, the agent holds wheat in its off hand to cause the sheep to approach it. The reward function also has the same structure as that task, albeit with different coefficients: $+|\Delta d_{\min}|$ for decreasing the minimum distance to the sheep and $+10$ for shearing it.

*Combat zombie*: Same as *combat spider*, but the enemy is a zombie. We increase the episode length to 1000, as the zombie has more health points than the spider.

*Combat enderman*: Same as *combat spider*, but the enemy is an Enderman. As with combat zombie, we increase the episode length to 1000. Note that Endermen are non-hostile (until directly looked at for sufficiently long or attacked) and have significantly more health points than other enemies. We thus enchant the agent's sword to deal more damage and decrease the initial spawn distance of the enderman from the agent.

*Combat pigman*: Same as *combat spider*, but the enemy is a hostile zombie pigman. As with combat zombie, we increase the episode length to 1000.

### B.2 POLICY AND TRAINING DETAILS

For our actual RL algorithm, we use the Stable-Baselines3 (version 2.0.0) implementation of clipping-based PPO (Raffin et al., 2021), with hyperparameters presented in Table 6. Many of these parameters are the same as the ones presented by Fan et al. (2022). For the spider trials, we use a cosine learning rate schedule:

$$\text{LR}(\text{current train step}) = \text{Min LR} + (\text{Max LR} - \text{Min LR}) \left( \frac{1 + \cos\left( \pi \frac{\text{current train step}}{\text{total train steps}} \right)}{2} \right) \quad (1)$$

We also present the policy and VLM hyperparameters in Table 7. The hyperparameters and architecture of the MLP part of the policy are primarily defined by the default values and structure defined by the Stable-Baselines3 `ActorCriticPolicy` class. Note that the no generation ablation, VLM image encoder baseline, and MineCLIP trials do not generate text with the VLM, and so all do not use the associated process's hyperparameters. The MineCLIP trials also do not use a Transformer layer in the policy, due to not receiving token sequence embeddings. It instead just uses a MLP, but with two hidden layers (to supplement the lowered policy expressivity due to the lack of a Transformer layer).

Additionally, InstructBLIP's token embeddings are larger than ViT-g/14's (used in the VLM image encoder baseline), and so may carry more information. However, the VLM does not receive any privileged information over the image encoder *from the task environment* – any additional information

| Policy Transformer Hyperparameters | |
|---|---|
| Transformer Token Size | 512 / 128 |
| Transformer Feedforward Dim | 512 / 128 |
| Transformer Number Heads | 2 |
| Transformer Number Decoder Layers | 1 |
| Transformer Number Encoder Layers | 1 |
| Transformer Output Dim | 128 |
| Transformer Dropout | 0.1 |
| Transformer Nonlinearity | ReLU |

| Policy MLP Hyperparameters | |
|---|---|
| Number Hidden Layers | 1 |
| Hidden Layer Size | 128 |
| Activation Function | tanh |

| VLM Generation Hyperparameters | |
|---|---|
| Max Tokens Generated | 6 |
| Min Tokens Generated | 6 |
| Decoding Scheme | Greedy |

Table 7: All policy hyperparameters for all Minecraft tasks. Smaller token sizes and feedforward dimensions are used for *combat [zombie/enderman/pigman]*.

in the VLM's representations is therefore purely from the model's prompted internal knowledge. Still, to ensure consistent policy expressivity, we include a learned linear layer projecting all representations for this baseline and our approach to the same size (512 dimensions) so that the rest of the policy is the same for both.

Minecraft training runs were run on 16 A5000 GPUs (to accommodate the 16 seeds).

## C   HABITAT OBJECTNAV DETAILS

### C.1   ENVIRONMENT DETAILS

The spaces and agent/task specifications are largely the same as the defaults provided by Habitat, as specified in the HM3D ObjectNav configuration file (Savva et al., 2019).

**Spaces.** The observation space for Habitat consists of the following:

1. **RGB:** Egocentric RGB images from the agent. (480, 640, 3)-size tensor of integers $\in \{0, 1, ..., 255\}$. By default, agents also receive depth images, but we remove them to ensure that state representations are grounded primarily in visual observations.

2. **Position:** Horizontal Cartesian coordinates of agent. 2-element vector of floats.

3. **Compass:** Yaw of the agent. Single floats.

4. **Previous Action:** The previous action taken by the agent. Set to no operation at the start of each episode. One-hot vector of size $|\mathcal{A}| = 4$.

5. **Object Goal:** Which object the agent is aiming to find. One-hot vector of size 3.

The action space is the standard Habitat-Lab action space, though we remove the pitch-changing actions, leaving only four:

1. **Turn:** Turn left or right, changing the yaw by $30°$.

2. **Move Forward:** Move forward a fixed amount or until the agent collides with something.

3. **Stop:** Ends the episode, indicating that the agent believes it has found the goal object.

All observations, actions, and associated dynamics are deterministic.

**World specifications.** In ObjectNav, an agent is spawned in a household environment and must find and navigate to an instance of a specified target object in as efficient a path as possible. Doing so

effectively requires a common-sense understanding of where household objects are often found and the structure of standard homes.

Habitat provides a standardized train-validation split, consisting of 80 household scenes for training (from which one can run online RL or collect data for offline RL or BC) and 20 novel scenes for validation, thereby testing policies' generalization capabilities. These scenes come from the Habitat-Matterport 3D v1 dataset (Ramakrishnan et al., 2021).

### C.2 Policy and Training Details

In line with previous work (Ramrakhya et al., 2023; Yadav et al., 2023b; Majumdar et al., 2023), we train our policies with behavior cloning (BC) on the Habitat-Web human demonstration dataset of 77k trajectories (12M steps) (Ramrakhya et al., 2022). We adopt many of the same design choices provided by said prior works, but with a few critical differences:

1. Due to compute limitations, we were unable to train on the full dataset (as those original works used 512 parallel environments to roll out demo trajectories and collect data). Instead, we used a subset of the dataset, built by dividing the dataset by both target object and scene, then sampling every tenth demo. This would ensure that our training data still contained examples from every training scene + target object combination that existed. In total, our subsampled dataset contains approximately 1.1M steps over 7550 trajectories.

2. We adopt the same optimizer, scheduler, and associated hyperparameters as Majumdar et al. (2023), but find a learning rate of $1e - 4$ to be more effective than their $1e - 3$.

3. Rather than sampling partial trajectory rollouts from 512 parallel environments as done by Majumdar et al. (2023), our batches contain full trajectories, though with the same total number of transitions per batch as in that work. This means that our batches potentially contain less diverse data (due to observations from fewer different total scenes being present), but allow us to compute up-to-date full trajectory hidden states for the RNN portion of our policy. We use gradient accumulation to achieve this, once again due to compute limitations.

4. While Majumdar et al. (2023) trains for 24k gradient steps (observing approximately 400M transitions.), we find using only approximately a tenth of that (40 epochs through our smaller dataset, so around 40M transitions) to reach peak performance for our policy. The scheduler still assumes the full training run will last for 400M transitions, so our LR decays at the same rate as with VC-1. Furthermore, for fairness, we leave our VC-1 baseline policies (trained on our subsampled datasets) training beyond 40 epochs, and report their validation performance at both 40 and 120 epochs (when its performance saturates).

5. For policies that receive visual observations as a sequence of tokens (PR2L, VC-1 with patch embeddings), we apply 2D average pooling with kernel sizes of $4 \times 4$ to reduce down to 16 tokens. Then, we pass those tokens through a learned Transformer layer, instead of the learned compression layer used by Majumdar et al. (2023). We do this to ensure that policy performance differences are due to representation quality, not architecture.

6. We employ inflection upweighting during training, as done by Ramrakhya et al. (2023); Yadav et al. (2023b); Majumdar et al. (2023). However, we also categorically upweight the cross entropy loss of stopping and turning by $1.5$ (due to them being uncommon but important), as we observe this increases learning speed for all policies.

7. We do not employ any image augmentation or loss regularization to prevent overfitting. However, we find our policy exhibits strong generalization performance in unseen validation scenes nonetheless.

For PR2L-specific design choices:

1. Our chosen VLM is the Prismatic VLM (Karamcheti et al., 2024) with Dino+SigLIP as a vision backbone and Llama2-7B-pure as the language backbone. We use the 224px version, which maps images to 256 visual tokens (which, as described above, get compressed into 16 via pooling).

2. To reduce the size of VLM representations for PR2L, we embed one observation (sampled uniformly at random) from each trajectory in our subsampled dataset with our VLM, then

compute all resulting tokens' principle component vectors. We then use said vectors to lower all tokens' dimensionality down from 4096 to 1024 (i.e., corresponding approximately to their first 1024 principle components).

3. Like with the Minecraft experiments, we take the VLM's last two layers' embeddings and treat them as our promptable representations. However, unlike with Minecraft, we stack each VLM token's two embeddings (forming new embeddings of size 2048), rather than concatenate all of them.

4. For generating text in response to our task-relevant prompt, we use sample-based decoding with fixed random seed prior to the decoding with temperature $0.4$ and $32 - 48$ new tokens generated.

5. The learned Transformer layer of our policy is the same as the one used in the Minecraft experiments, but with token embedding sizes of 1024.

All Habitat training was done on an A100 GPU server. Generation of data and evaluations were done on 16 A5000 GPUs for parallelization.

## D  SIMPLIFIED HABITAT OFFLINE RL EXPERIMENTS

While our primary Habitat experiments use behavior cloning to stay consistent with past works, we also run offline RL experiments on a simplified version of ObjectNav to better explore how VLM representations aid action learning. We discuss the details of said setting now.

### D.1  ENVIRONMENT DETAILS

We pick 32 reconstructed 3D home environments with at least one instance of each of the three target objects (toilet, bed, and sofa) and an annotator quality score of at least 4 out of 5. We choose to remove *plants* and *televisions* from the goal object set due to finding numerous unlabeled instances of them. Additionally, we remove chairs, as they are significantly more common than other goal objects and thus usually can be found in much shorter episodes. This simplified problem formulation enables us to remove many of the "tricks" that aid ObjectNav, such as using omnidirectional views or policies with history; our agent makes action decisions purely based on its current visual observation and pose, allowing us to do "vanilla" RL to better isolate the effect of PR2L.

To generate data, we use Habitat's built-in greedy shortest geodesic path follower. Imitating such demonstrations allows policies to learn unintuitively emergent and performant navigation behaviors (Ehsani et al., 2023) at scale. For each defined starting location in our considered households, we autonomously collect data by using the path follower to navigate to each reachable instance of the corresponding goal object. This yields high quality, near-optimal data. We then supplement our dataset by generating lower-quality data. Specifically, for each computed near-optimal path from a starting location to a goal object instance, we choose to inject action noise partway through the trajectory (uniformly at random from $0 - 90\%$ of the way through). At that point, all subsequent actions have a $0 - 50\%$ probability (again chosen uniformly at random) of being a random action other than the one specified by the path follower. To ensure that paths are sufficiently long, we choose to make the probability of choosing the stop action $10\%$ and the other two movement actions $45\%$. In total, we collect 107518 observations over 2364 trajectories.

**Reward functions.** The ObjectNav challenge evaluates agents based on the average "success weighted by path length" (SPL) metric (Yadav et al., 2023a): if an agent succeeds at taking the *stop* action while close to an instance of the goal object, it gets $SPL(p, l) = \frac{l}{\max(l,p)}$ points, where $l$ is the actual shortest path from the starting point to an instance of the goal object and $p$ is the length of the path that the agent actually took during that particular episode. If the agent stops while not close to the target object, the SPL is 0. Thus, taking the most efficient path to the nearest goal object and stopping yields a maximum SPL of 1.

We use this to design our reward function. Specifically, when the agent stops, it receives a reward of $+10 SPL(p, l)$. Additionally, we add a shaping reward of the change in geodesic distance to the nearest goal object instance each time the agent moves (where lowering that distance yields a positive reward).

### D.2 Policy and Training Details

For our offline RL experiments in Habitat, we use Conservative Q-Learning (CQL) on top of the Stable-Baslines3 Contrib codebase's implementation of Quantile Regression DQN (QR-DQN) Kumar et al. (2020); Dabney et al. (2017). We choose to multiply the QR-DQN component of the CQL loss by $0.2$. Using the notation proposed by Kumar et al. (2020), this is equivalent to $\alpha = 5$, which said work also uses. Other hyperparameters are $\tau = 1$, $\gamma = 0.99$, fixed learning rate of $1e - 4$, 100 epochs, and 50 quantiles (no exploration hyperparameters are specified, since we do not generate any new online data).

The policy architecture used for Habitat experiments are the same as those used for PPO, though the final network outputs quantile Q-values for each action (rather than just a distribution over actions). The action with the highest mean quantile value is chosen at evaluation time.

During training, we shuffle the data and load full offline trajectories until the buffer has at least $32 \times 1024 = 32768$ transitions or all trajectories have been loaded once that epoch. We then uniformly sample and train on batches of size 512 transitions from the buffer until each transition has been trained on once in expectation (e.g., $\sim \frac{\text{number of transitions in the buffer}}{512}$ batches). Each batch is used for 8 gradient steps before the next is sampled. We choose this data loading scheme to fit the training infrastructure provided by Stable-Baselines3 while not using up too much memory at once.

### D.3 Experiments and Results

Our primary comparison is once again between our promptable representations and general-purpose non-promptable ones. We thus repeat the baseline described previously for Minecraft in Section 4.1, training a single agent for all three ObjectNav tasks using both PR2L and the VLM image encoder representations. We empirically note that longer visual embedding sequences tend to perform better in Habitat. To control for this, we opt to use InstructBLIP's Q-Former unprompted embeddings instead of the ViT embeddings directly (which are much longer than PR2L's embedding sequences). As InstructBLIP uses the former representations to extract visual features to be projected into language embedding space, this serves to close the gap in embedding sequence length between our two conditions while still providing us with general visual features that the VLM processes via prompting. In this case, we use the same InstructBLIP model as the Minecraft experiments and choose "What room is this?" as our task-relevant prompt.

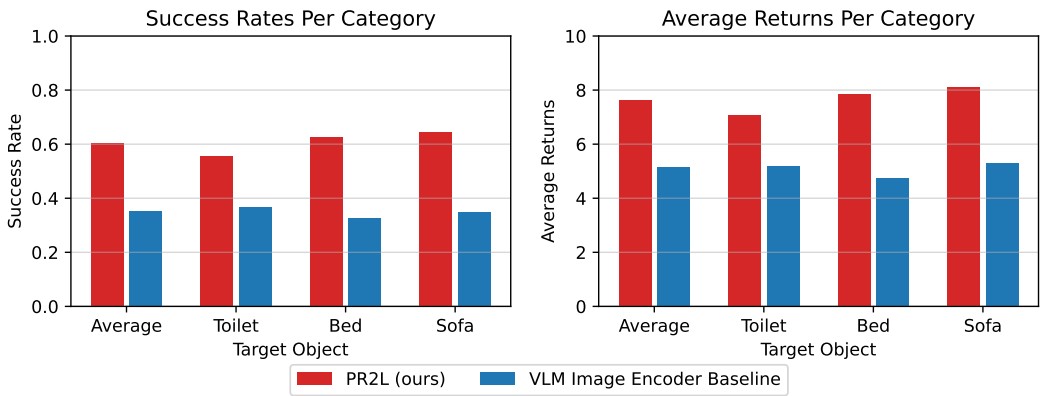

Figure 5: **Offline RL performance of PR2L and baselines in Habitat ObjectNav**. Plots show final evaluation success rates and average returns per target object and overall. PR2L outperforms the baseline in all cases.

We report evaluation success rates and average returns for the simplified Habitat ObjectNav setting in Figure 5. PR2L achieves nearly double the average success rate of the baseline ($60.4\%$ vs. $35.2\%$), supporting the hypothesis that PR2L works especially well when exploration is not needed. Lastly, in Appendix H.2, we find that PR2L causes the VLM to produce highly structured representations that correlate with an expert policy's value function: high-value states are typically labeled by the VLM as being from a room where one would expect to find the target object.

| Task | PR2L (Ours) | VLM Image Encoder | Ablations | | | |
|------|-------------|-------------------|-----------|-----|-----|-----|
| | | | No Prompt | No Generation | Change Aux. Text | Oracle Detector |
| *Combat Spider* | **97.6 ± 14.9** | 51.2 ± 9.3 | 72.6 ± 14.2 | 66.6 ± 11.8 | 80.1 ± 12.6 | 58.0 ± 13.4 |
| *Milk Cow* | **223.4 ± 35.4** | 95.2 ± 18.7 | 116.6 ± 25.9 | 160.2 ± 23.6 | 80.5 ± 17.8 | 178.4 ± 42.5 |
| *Shear Sheep* | **37.0 ± 4.4** | 23.0 ± 3.6 | 23.8 ± 3.2 | 26.1 ± 4.5 | 27.8 ± 4.6 | 27.4 ± 9.3 |

Table 8: Minecraft ablations, VLM image encoder baseline, and our full approach. All achieve worse performance than PR2L. Values are final IQM success counts and intervals are the standard error.

# E EXTENDED DISCUSSION OF TASKS AND RESULTS

## E.1 NOTES ON TASK-SPECIFIC SYSTEMS

We designed experiments to specifically investigate the use of VLM embeddings as task-specific promptable representations for downstream sensorimotor policy learning. As such, we compare with other works that propose or evaluate either learning from scratch or from pre-trained representations, but *not* to systems in Minecraft and Habitat that require domain-specific engineered systems beyond just policy learning (such as Luo et al. (2022); Zhu et al. (2022)) or which target learning or producing higher-level plans or abstractions (such as Wang et al. (2023b)).

Such comparisons are not made as these works either aim to investigate other problems in control or are aiming to develop highly specialized and task-specific systems (whereas we present a general approach for policy learning). For instance, Voyager shows how an LLM can reason about and compose high-level hand-crafted control primitives (Wang et al., 2023a). Voyager's ability to complete harder tasks comes from its access to powerful hand-crafted high-level primitives that extensively leverage oracle information, which are composed into skills by GPT-4 (which does not handle any low-level control). Said hand-coded control primitives used in Voyager are very advanced and do much of the heavy-lifting. In particular, Voyager gives GPT-4 access to a dedicated `killMob(<entity name>)` control primitive function. This function calls a separate `bot.pvp.attack(<entity>)` (hand-written) function, which calls a hard-coded oracle pathfinder, aiming controller, and attack function to repeatedly approach and attack the specified entity until it is defeated. Thus, for Voyager, the skill for hunting sheep simply fills in the powerful `killMob()` primitive function with "sheep" as the target, abstracting away all low-level control via the oracle hand-written controllers.

Vitally, unlike PR2L, Voyager does not investigate how to use (V)LMs to learn these primitives. It thus cannot be applied to settings that lack such primitives (e.g., because oracle path planners are not available, like in Habitat). This makes PR2L complementary: we directly learn a policy to link observations to low-level actions (turning, moving, attacking, etc) via RL with no oracle information, while Voyager aims to compose pre-existing primitives into skills via LLMs.

## E.2 NOTES ON DREAMER V3

We note that PR2L just proposes to use VLMs as a source of task-specific representations for RL tasks; it does not prescribe which learning algorithm to use. Therefore, in principle, one could use Dreamer in conjunction with PR2L and gain benefits from both the VLM representation and the choice of a strong model-based RL algorithm. However, while we leave this to future works, our Minecraft comparison (c) measures how well the approach does on our Minecraft tasks (as the original paper focuses more on the component subtasks involved in the *find diamond* task, all of which do not involve interacting with moving entities).

We find that Dreamer v3 is unable to learn our six tasks given the same number of environment interactions that PR2L+PPO was trained on. We hypothesize that this is due to its visual reconstruction-based world model not being suited for tasks requiring interaction with partially-observable, non-stationary autonomous entities (which all our tasks involve). We note that the last two rows of the figure visualizing model reconstructions in the original Dreamer v3 paper shows that its world model fails to reconstruct an observed pig (Hafner et al., 2023), supporting our hypothesis. This highlights the need for robust representations that are conducive to world model learning, with PR2L's capabilities to elicit task-relevant visual semantic features via prompting being one possibility for doing so.

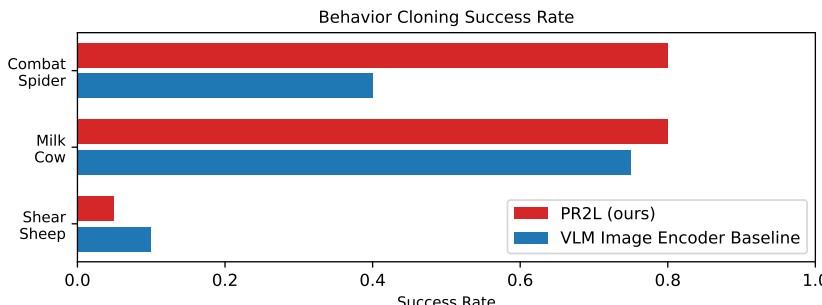

Figure 6: Success rates for BC on either PR2L or VLM image encoder baseline representations for all original tasks. PR2L excels at *combat spider*, even after the policy is trained for a single epoch.

## F   ABLATIONS

We run four ablations on *combat spider*, *milk cow*, and *shear sheep* to isolate and understand the importance of various components of PR2L. First, we run PR2L with *no prompt* to see if prompting with task context actually tailors the VLM's generated representations favorably towards the target task, improving over an unprompted VLM. Note that this is not the same as just using the image encoder (comparison (a)), as this ablation still decodes through the VLM, just with an empty prompt. Second, we run PR2L with our chosen prompt, but *no generation* of text – i.e., the policy only receives the embeddings associated with the image and prompt (the left and middle red groupings of tokens in Figure 2, but not the right-most group). This tests the hypothesis that representations of generated text might make certain task-relevant features more salient: e.g., the embeddings for "Is there a cow in this image?", might not encode the presence of a cow as clearly as if the VLM generates "Yes" in response, impacting downstream performance. Third, to check if our prompt evaluation strategy provides a good proxy for downstream task performance while tuning prompts for P2RL, we run PR2L with alternative prompts that were not predicted to be the best, as per our criterion in Appendix A. We thus remove the auxiliary text from the prompt for *combat spider* and add it for *milk cow* and *shear sheep*. Lastly, to see if PR2L embeddings are just better due to them encoding entity detection, we train a VLM image encoder policy with an additional ground truth oracle target entity detector as a feature.

Results from these additional experiments are presented in Table 8. In general, all ablations perform worse than PR2L. For *milk cow*, we note the most performant ablation is no generation, perhaps because the generated text is often wrong; among the chosen prompts, it yields the lowest true positive and negative rates for classifying the presence of its corresponding target entity (see Table 5 in Appendix A), though adding auxiliary text makes it even worse, perhaps explaining why *milk cow* experienced the largest performance decrease from adding it back in. Based on these overall trends, we conclude that (i) the *promptable* and *generative* aspects of VLM representations are important for extracting good features for control tasks and (ii) our simple evaluation scheme is an effective proxy measure of how good a prompt is for PR2L.

## G   MINECRAFT BEHAVIOR CLONING EXPERIMENTS

We collected expert policy data by training a policy on MineCLIP embeddings to completion on all of our original tasks and saving all transitions to create an offline dataset. We then embedded all transitions with either PR2L or the VLM image encoder. Finally, we train policies with behavior cloning (BC) on successful trajectories under a specified length (300 for *combat spider*, 250 for *milk cow*, and 500 for *shear sheep*) from either set of embeddings for all three tasks, then evaluate their task success rates.

Results are presented in Figure 6. We first note that, since the expert data was collected from a policy trained on MineCLIP embeddings, the *shear sheep* policy is not very effective (as we found in Table 2). Both resulting *shear sheep* BC policies are likewise not very performant. We find that *combat spider* in particular shows a very large gap in performance: the PR2L agent achieves approximately

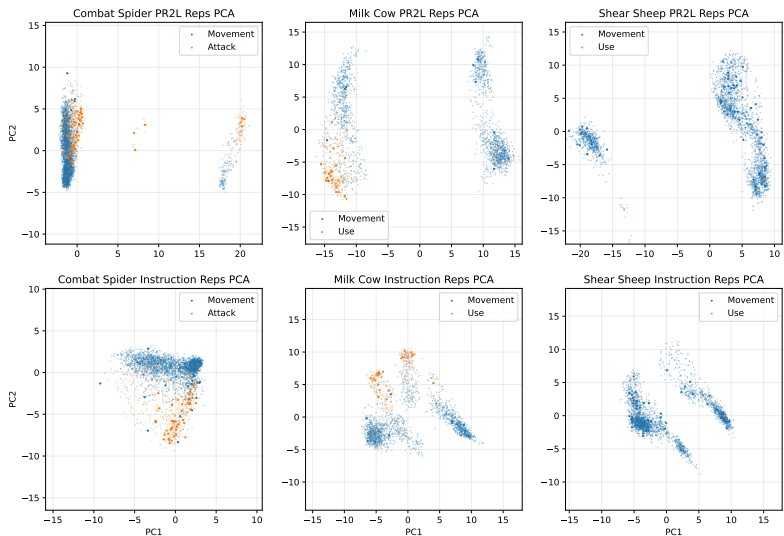

Figure 7: **PCA of PR2L representations of observations from twenty episode rollouts of expert policies in all three Minecraft tasks.** Larger points correspond to transitions where the expert received > 0.1 reward. We vary the prompt to be either our task-relevant prompt or the RT-2-style baseline instruction prompt. Our prompt's representations are bi-modal, with the clusters on the left corresponding to the VLM outputting "yes" (the entity is in view). We find that most functional actions (orange points) that yielded rewards are located in said clusters. Note that, since these expert policies are trained on top of MineCLIP embeddings, the *shear sheep* policy is not very performant, as seen in Table 2.

twice the success rate of the VLM image encoder agent *after training for just a single epoch*. The comparatively small amount of training and data necessary to achieve near-expert performance for this task supports our hypothesis that promptable representations from general-purpose VLMs do not help with exploration (they work better in offline cases, where exploration is not a problem), but instead are particularly conducive to being linked to appropriate actions even though the VLM is not producing actions itself. Further investigation of this hypothesis is presented in Appendix H.

## H    REPRESENTATION ANALYSIS

Why do our prompts yield higher performance than one asking for actions or instruction-following? Intuitively, despite appropriate responses to our task-relevant prompts not directly encoding actions, there should be a strong correlation: e.g., when fighting a spider, if the spider is in view and the VLM detects this, then a good policy should know to attack to get rewards. We therefore wish to investigate if our representations are conducive to easily deciding when certain rewarding actions would be appropriate for a given task – if it is, then such a policy may be more easily learned by RL, which would explain PR2L's improved performance over the baselines.

### H.1    MINECRAFT ANALYSIS

To investigate this, we use the embeddings of our offline data from the BC experiments (collected by training a MineCLIP encoder policy to high performance on all of our original three tasks, as discussed in Appendix G). We specifically look at the embeddings produced by a VLM when given our standard task-relevant prompts and when given the instructions used for our RT-2-style baseline. We then perform principal component analysis (PCA) on the tokenwise average of all embeddings for each observation, thereby projecting the embeddings to a 2D space with maximum variance.

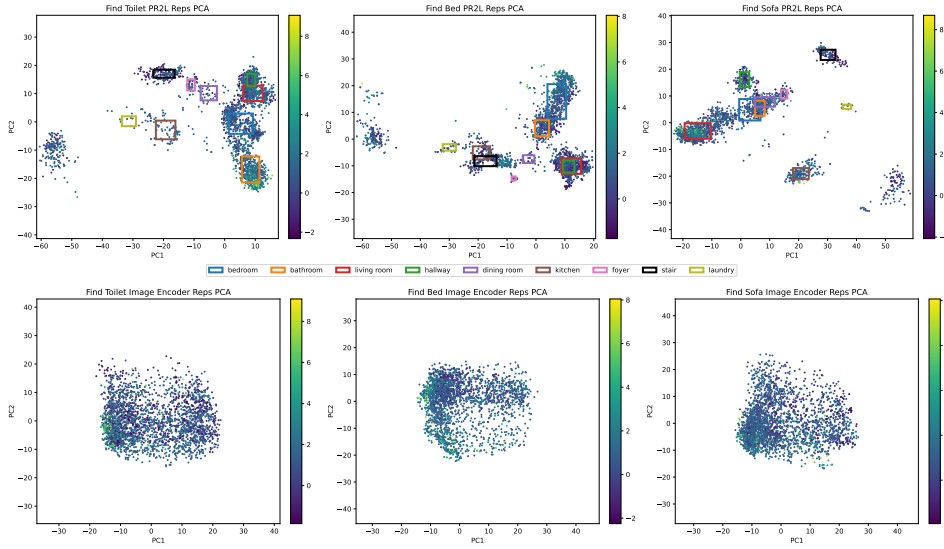

Figure 8: **PCA of PR2L (above) and image encoder (below) representations of observations from thirty episode rollouts of expert policies in all Habitat tasks.** The points' colors correspond to their value under Habitat's built-in oracle shortest path follower (a near-optimal policy). More yellow is better. Boxes correspond to points the VLM has labeled as a given household room, in response to the task prompt of "What room is this?" This analysis aligns with intuition: for *find toilet*, high value observations tend to be labeled as bathrooms (orange box), *find bed*'s tend to be labeled as bedrooms (blue), and *find sofa*'s are labeled as living rooms (red).

We visualize these low-dimensional space in Figure 7 for the final 20 successful observations from each task, with the point colors of orange and blue respectively indicating whether the observation results in a functional action (attack or use item) or movement (translation or rotation) by the expert policy. Additionally, we enlarge points corresponding to when the agent received rewards in order to recognize which actions aided in or achieved the task objective.

We find that our considered prompts resulted in a bimodal distribution over representations, wherein the left-side cluster corresponds to the VLM outputting "yes (the entity is in view)" and the right-side one corresponds to "no." Additionally, observations resulting in functional actions that received rewards (large orange points in Figure 7) tend to be on the left-side ("yes") cluster for representations elicited by our prompt, but are more widely distributed in the instruction prompt case, in agreement with intuition. This is especially clear in the *milk cow* plot, wherein nearly all rewarding functional actions (using the bucket on the cow to successfully collect milk) are in the lower left corner.

This analysis supports that the representations yielded by InstructBLIP in response to our chosen style of prompts are more structured than representations from instructions. Such structure is useful in identifying and learning rewarding actions, even when said actions were taken from an expert policy trained on unrelated embeddings. This suggests that such representations may similarly be more conducive to being mapped to good actions via RL, which we observe empirically (as our prompt's representations yield more performant policies than the instructions for the RT-2-style baseline).

## H.2 HABITAT ANALYSIS

Likewise, we conduct a similar analysis on the Habitat data from our simplified setting. Specifically, we wish to see if PR2L produces representations that are conducive to extracting the *value function* of a good policy. Since the chosen Habitat ObjectNav prompt is "What room is this?" we expect the state representations to be clustered based on room categories. Intuitively, states corresponding to the room one is likely to find the target object should have the highest values.

As shown in Figure 8, we thus used PCA to project expert trajectories' PR2L and general image encoder state representations (generated with Habitat's geodesic shortest path follower) to two

dimensions, then colored each one based on their value under said near-optimal policy. We also plotted the mean and standard deviation of all points labeled as each room, visualizing them as axis-aligned bounding boxes. Note that each upper subplot in Figure 8 has a cluster of points far from all boxes. These correspond to the VLM generating nothing or garbage data with no room label.

This visualization qualitatively agrees with intuition. High value states tend to be grouped with the room the corresponding target object is often found in: *find toilet* corresponds to bathrooms, *find bed* to bedrooms, and *find sofa* to living rooms. Comparatively, the general image encoder features do not have such semantically meaningful groupings; all observations are clustered together and, within that single grouping, high-value observations are more spread out. This all supports the idea that prompting allows representations to take on structures that correlate well to value functions of good policies.

## I  CODE SNIPPETS

We provide some code snippets showcasing instantiations of PR2L.

```python
class Policy(torch.nn.Module):
    def __init__(self, num_actions, tf_embed_dim=4096):
        """Policy that accepts promptable reps as input"""
        super().__init__()
        # Project down VLM embed dimensions
        self.embed_fc = torch.nn.Linear(tf_embed_dim, 1024)
        # Predict actions
        self.action_fc = torch.nn.Linear(1024, num_actions)
        # Transformer layer to condense promptable reps to 1 token
        self.transformer = torch.nn.Transformer(
            1024,
            1,
            num_encoder_layers=1,
            num_decoder_layers=1,
            dim_feedforward=1024,
            batch_first=True,
        )
        self.cls = torch.nn.Embedding(1, 1024)  # cls tokens

    def forward(self, x):
        seq, mask = x
        bs, traj_len, num_tokens, _ = seq.shape

        # [batch*traj_len, num tokens, token size]
        seq = seq.reshape(bs * traj_len, num_tokens, -1)
        # [batch*traj_len, num tokens]
        mask = mask.reshape(bs * traj_len, num_tokens)

        # Project down
        # [batch*traj_len, num tokens, tf dim]
        seq = self.embed_fc(seq)

        # Get CLS embedding
        cls = self.cls(torch.zeros([bs * traj_len, 1],
            device=seq.device, dtype=int))

        # Get summary embedding
        # [batch*traj_len, 1, tf dim]
        cls_embed = self.transformer(
            seq,  # Encoder input
            cls,  # Decoder input
            # Apply mask
            src_key_padding_mask=mask,
            memory_key_padding_mask=mask,
        )
```

```
        # [batch, traj_len, d_model]
        cls_embed = cls_embed.reshape(bs, traj_len, -1)

        # Predict actions
        # [batch, traj_len, actions]
        return self.action_fc(cls_embed)
```

Listing 1: Example policy for PR2L.

```python
def process_obs(model, processor, image, prompt, device, last_n=2):
    inputs = processor(images=image, text=prompt, return_tensors="pt").to
    (device)

    # Generate text in response to prompt and extract embeddings
    outputs = model.generate(
        **inputs,
        output_hidden_states=True,
        return_dict_in_generate=True,
        # Any other generation parameters (min/max tokens, temp, etc)
    )
    hs = outputs["hidden_states"]

    # Get image and prompt token embeds
    # Any additional processing should happen here (eg pooling of visual
    tokens)
    # [last_n, num img + prompt tokens, tf_embed_dim]
    image_and_prompt_embs = torch.cat(hs[0], dim=0)[-last_n:]

    # Get decoded token embeds
    # [last_n, num decoded tokens, tf_embed_dim]
    dec_embs = []
    for dec_hs in hs[1:]:
        # [last_n, 1, tf_embed_dim]
        dec_hs = torch.cat(dec_hs, dim=0)[-last_n:]
        dec_embs.append(dec_hs)
    # [last_n, num decoded tokens, tf_embed_dim]
    dec_embs = torch.cat(dec_embs, dim=1)

    # [last_n, num total tokens]
    seq_embs = torch.cat([image_and_prompt_embs, dec_embs], dim=1)
    tf_embed_dim = seq_embs.shape[-1]

    # [bs=1, seq_len=1, last_n*num total tokens, tf_embed_dim]
    seq_embs = seq_embs.reshape(1, 1, -1, tf_embed_dim)

    mask = torch.zeros(seq_embs[:-1], type=int)

    return seq_embs, mask
```

Listing 2: Example code for extracting promptable representations from a VLM.

```python
# Create VLM and processor (InstructBLIP, for example)
model = InstructBlipForConditionalGeneration.from_pretrained(
    "Salesforce/instructblip-vicuna-7b"
)
processor = InstructBlipProcessor.from_pretrained("Salesforce/
    instructblip-vicuna-7b")

# Set device, can also change dtype if desired
device = "cuda:0"
model = model.to(device)

# Create env
env = ...
```

```
# Create policy. This can be trained via RL or BC as needed.
policy = Policy(env.num_actions).to(device)

# Define task-relevant prompt
prompt = "Would a toilet be found here? Why or why not?"

# To predict an action, get an RGB obs from the env and process it with
    the VLM
obs = env.reset()
seq, mask = process_obs(model, processor, obs, prompt, device)

# Then, pass it through the policy to get action logits and step env
act_logits = policy.forward((seq, mask)).reshape(env.num_actions)
action = torch.argmax(act_logits)
obs, _, _, _ = env.step(action)
```
Listing 3: Example usage of the above function and policy.

## J   EXTENDED LITERATURE REVIEW

**Learning in Minecraft.** We now consider some current approaches for creating autonomous learning systems for tasks in Minecraft. Such works highlight some of the difficulties prevalent in tasks in said environment. For instance, since Minecraft tasks take place in a dynamic open world, it can be difficult for an agent to determine what goal it is attempting to reach and how close it is to reaching that goal. Cai et al. (2023) tackles these issues by introducing and integrating a training scheme for self-supervised goal-conditioned representations and a horizon predictor. Zhou et al. (2023) learns a model from visual observations to discriminate between expert state sequences and non-expert ones, which provides a source of intrinsic rewards for downstream RL tasks (as it pushes the policy to learn to match the expert state distribution, which tend to be "good" states for accomplishing tasks in Minecraft).

**Foundation Models and Minecraft.** Likewise, there has been much interest in applying foundation models – especially (V)LMs – to Minecraft tasks. Baker et al. (2022) pretrains on large scale videos, which enabled the first agent that could learn to acquire diamond tools (thereby completing a longstanding challenge in the MineRL competition Kanervisto et al. (2022)). LMs have subsequently also been used to produce graphs of proposed skills to learn or technology tree advancements to make in the form of structured language (Nottingham et al., 2023; Zhu et al., 2023; Yuan et al., 2023; Wang et al., 2023b). Other works propose to use the LLM to generate actions or code submodules given textual descriptions of observations or agent states (Wang et al., 2023a). Finally, VLMs have been used largely for language-conditioned reward shaping (Fan et al., 2022; Ding et al., 2023). In contrast, we use VLMs as a source of representations for learning of atomic tasks (as defined by Lin et al. (2023a)) that have pre-defined reward functions; the latter works can thus be used in conjunction with our proposed approach for tasks where these vision-language reward functions are appropriate.

