# OpenReview forum: "Vision-Language Models Provide Promptable Representations for Reinforcement Learning"
_ICLR.cc/2025/Conference — ICLR 2025 Conference Withdrawn Submission_

### Official Review · Reviewer_yoee · 2024-11-01

**Soundness:** 3
**Presentation:** 3
**Contribution:** 2
**Rating:** 5
**Confidence:** 5

**Summary:**

This paper proposes an approach that uses the vision-language models (VLMs) pre-trained on Internet-scale data for embodied RL.
They initialize the embedding module with VLMs by using them to generate world knowledge representations.
Specifically, the embeddings encode semantic features of visual observations based on the VLM’s internal knowledge and reasoning capabilities, as elicited through prompts that provide task context and auxiliary information.
They evaluate their method on visually complex, long-horizon RL tasks in Minecraft and robot navigation in Habitat and outperform equivalent policies trained by other methods.

**Strengths:**

This paper asks VLM questions about observations that are related to the given control task, priming it to attend to task-relevant features.
In this method, the model yields task-specific features capturing information that is particularly conducive to learning a considered task.

**Weaknesses:**

1. The paper proposes inspiring task-relevant representations by posing questions to vision-language models (VLM) based on visual observations. While this approach is reasonable, it is already quite common in the use of large models.
In line 116, the authors claim that their prompts do not rely on target prior information. However, in the experimental sections, particularly in the navigation and Minecraft tasks, the prompts appear to incorporate task priors. For example, in Figure 2, the navigation task is to "find toilet," and the prompt used is “Would a toilet be found here?” This prompt clearly includes task-specific prior information.
Could you clarify the definition of "target prior information" and how it differs from the task-specific information included in their prompts?

2.The construction of a reasonable prompt has great importance to the method's performance. However, the authors do not provide a universal method for prompt construction for different tasks. For example, the authors propose different prompt templates for navigation tasks and Minecraft tasks. Could the authors provide detailed rules for building prompt templates for different tasks?


3.The encoded representations from different layers of the VLM have varying impacts on downstream tasks. Could authors provide additional ablation studies to verify whether the semantic information in the encoded representations increases with the number of layers? Need more analysis is needed to justify the choice of using representations from the final layers.

4.The policy module in this method requires specific learning on downstream tasks. Could the authors discuss whether fine-tuning the VLM on downstream tasks, in addition to using prompts to inspire representations, would be more effective?

**Questions:**

1. Could the authors emphasize the primary differences between the proposed approach and previous methods regarding the use of prior-based prompts, highlighting the innovation in this work?

2. Could the authors provide detailed guidelines on how prompts are specifically tailored for downstream tasks and clarify how different prompts affect representations? Could you provide a general framework or set of principles you used when designing prompts for different tasks. Examples of how they applied these principles to create prompts for specific tasks in Minecraft and Habitat. Guidelines on how other researchers could adapt their prompt design process to new tasks or domains

3. Could  the authors provide ablation studies to demonstrate the impact of representations from different VLM layers on model performance? Provide quantitative results showing how performance changes when using representations from different layers
Analyze and discuss any patterns observed in how semantic information changes across layers.

4. Could the authors also conduct experiments to test whether fine-tuning the VLM on downstream tasks would yield better results? The authors should discuss the trade-offs between their current approach and fine-tuning the VLM.
If feasible, conduct an experiment comparing their method to a fine-tuned VLM on a subset of tasks.
Analyze the computational requirements and potential scalability issues of fine-tuning large VLMs for specific tasks.

5. Could the authors provide more detailed experimental comparisons with baseline algorithms, including specifics such as the number of training steps for each algorithm and examples of prompts used?

---

### Official Review · Reviewer_rCMC · 2024-11-04

**Soundness:** 3
**Presentation:** 3
**Contribution:** 3
**Rating:** 6
**Confidence:** 3

**Summary:**

This paper introduces PR2L, a novel way to utilize the pre-trained VLM as the feature extractor for visual states. By prompting the VLM with task-relevant questions, PR2L then aggregates embeddings of the VLM's final layers by an encoder-decoder Transformer to obtain a final summarized embedding, which is used as the state representation for policy learning. The effectiveness of PR2L is demonstrated in Minecraft and Habitat domains.

**Strengths:**

- The paper is presented clearly and is easy to follow.
- The experiments show that PR2L gains strong performances in discrete control benchmarks such as Minecraft and Habitat.
- The analyses are conducted thoroughly to understand the effectiveness of prompt selection.

**Weaknesses:**

- One of the questions the paper aims to investigate in the experiments is sample efficiency. However, there are no learning curves for either domain, making it difficult to assess how sample-efficient PR2L is.
- While the paper introduces several design choices for PR2L (e.g., decoding scheme, layers for embedding aggregation, prompt designs), only the prompt design is ablated.
- Many of the baselines considered (e.g., RT-2, Dreamer, VC-1, R3M) are evaluated in continuous control domains in their original papers, so it is unclear if PR2L outperforms them in this setting. Could the authors provide some experiments in continuous domains such as locomotion (e.g., DMControl) or manipulation (e.g., MetaWorld)?

Minor:
- Line 226, "green in Figure 2" -> orange in Figure 2?

**Questions:**

- While the PR2L is better than other baselines, what is the cost of inference time of PR2L compared to them (e.g., R3M, MineCLIP, STEVE-1)?
- In the Minecraft ablation, the policy with Oracle entity detection performs worse than PR2L. Can you explain what other factors contribute to the effectiveness of the PR2L's features? Could the authors provide text output examples generated by the VLM in Minecraft to understand this aspect better?
- In Habitat, CoT helps significantly improve the quality of features compared to when CoT is not used. Does it have the same effect in Minecraft?
- PR2L uses a learned Transformer to aggregate embeddings and provide the final fixed-length embedding. Do you think using a sentence embedding model (e.g., SentenceBERT [1] or Universal Sentence Encoder [2]) also to provide a fixed-length embedding of decoded text would have a similar effect?

[1] Reimers N. Sentence-BERT: Sentence Embeddings using Siamese BERT-Networks. arXiv:1908.10084. 2019.
[2] Cer D. Universal sentence encoder. arXiv:1803.11175. 2018.

---

### Official Review · Reviewer_vkM7 · 2024-11-04

**Soundness:** 2
**Presentation:** 3
**Contribution:** 2
**Rating:** 3
**Confidence:** 4

**Summary:**

The main contributions of this paper are: (1) introducing Promptable Representations for Reinforcement Learning (PR2L), a method that leverages vision-language models (VLMs) to provide RL agents with task-specific semantic features through prompt-based embeddings, enhancing agents’ perceptual grounding and decision-making; (2) demonstrating that PR2L achieves better sample efficiency and task performance in complex, visually rich environments (e.g., Minecraft and Habitat) compared to non-prompted or baseline representations; and (3) showing that PR2L’s use of common-sense reasoning through chain-of-thought prompting improves generalization in unfamiliar environments without additional domain-specific fine-tuning.

**Strengths:**

Authors combine pre-trained vision-language models (VLMs) with reinforcement learning (RL) via prompt-based embeddings to leverage vast world knowledge in VLMs for contextualized agent learning. The results show PR2L’s performance gains over non-promotable and domain-specific embeddings across environments like Minecraft and Habitat. Well-structured explanations although some details on prompt design could be expanded for reproducibility.

**Weaknesses:**

While the paper presents an interesting integration of vision-language models (VLMs) with reinforcement learning (RL), it could benefit from more task-specific prompt design. Prompts are crafted based on heuristic task relevance, which may not fully optimize the embeddings for RL tasks. Extensive evaluation is required for prompt evaluation across diverse tasks to assess the representation's generalizability. Furthermore, the experimental evaluation, although thorough, is limited by the reliance on specific VLMs like InstructBLIP and Prismatic VLMs. Expanding the experiments to include various VLMs (BLIP3, LLaVA, etc) with varied training paradigms (e.g., contrastive VLMs like CLIP) would help to assess PR2L’s adaptability and robustness across models. Additionally, while the paper provides a comparison with baseline methods, further ablation studies—such as analyzing the impact of different VLM layers on task performance—could yield deeper insights into the mechanisms behind PR2L’s gains.

Moreover, the authors stated that “our method yields task-specific features capturing information particularly conducive to learning a considered task,” but this claim lacks clear empirical validation in the experimental results. Additionally, while the author mentioned that “the VLM does not just produce an un-grounded encoding of instructions, but embeddings containing semantic information relevant to the task, that is both grounded and informed by the VLM’s prior knowledge,” it is not clear how this grounding is achieved or measured.

Additional weakness:
- Limited ablation studies on the impact of different prompting strategies and VLM architectures. The paper primarily uses one VLM (InstructBLIP) without thoroughly comparing alternatives.
- The evaluation focuses heavily on gaming environments (Minecraft) and simulated navigation (Habitat), lacking real-world applications.
- The paper doesn't thoroughly address potential failure modes when VLM representations are misleading or incorrect
- Reproducibility issues, as the paper does not provide sufficient implementation details to replicate the experiments fully.

**Questions:**

- What evidence demonstrates that PR2L generates truly task-specific features? Can you provide a comparative analysis with baseline embeddings to validate this claim?
- How sensitive is performance to prompt engineering? What happens when VLM representations are incorrect?
- How exactly does PR2L achieve and measure the "grounding" of embeddings in the VLM's knowledge? What specific mechanisms ensure this?
- Have you explored automated or systematic approaches to prompt optimization rather than manual prompt design?
- Why limit testing to InstructBLIP and Prismatic? Why did we not consider recent VLM, such as LLaVABLIP-3
- Have you investigated which VLM layers provide the most effective embeddings for PR2L? An ablation study could be valuable here.
- Can PR2L's effectiveness be demonstrated beyond simulated environments? What about testing on real-world robotics tasks that require comprehending complex visual inputs?
Can we utilize a Knowledge base instead of a world model for these types of tasks to extract fine-grain contextual information?
- How does this compare to other methods of incorporating semantic knowledge into RL beyond the baselines tested?

---

### Note · Authors · 2024-11-21

I have read and agree with the venue's withdrawal policy on behalf of myself and my co-authors.